# Construction of Community Grid Unit Assessment System from the Perspective of Refined Governance

**Ningzhi Li [1] and Lilan Su [1,2,*]**

1. School of Public Administration, China University of Geosciences, Wuhan 430074, China; 20201004442@cug.edu.cn
2. Key Laboratory of The Ministry of Natural Resources for Research on Rule of Law, Wuhan 430074, China
* Correspondence: lilans@cug.edu.cn; Tel.: +86-136-0711-4864

**Abstract:** On the basis of reviewing the typical models of urban community grid governance, the framework of the community grid unit assessment system was put forward according to the main tasks of current grid management. The five grid governance tasks such as convenient service, risk reporting, contradiction persuasion, grid patrol, and problem investigation were taken as the first-level indicators, a theoretical model of a three-level grid unit assessment was established, and the community grid unit assessment system was constructed. Combined with big data platforms such as public security, civil administration, and the politics and law committee, this paper aims to solve the problems of performance assessment of community grid members, the distribution of grid members, the needs of the people, and the service of enterprises and volunteers, and promote the common development of communities, grid members, enterprises, and volunteers, and realize the refinement of grid governance.

**Keywords:** community governance; grid governance; refined governance; resource allocation





## 1. Introduction

The Third Plenary Session of the Eighteenth Central Committee of the Communist Party of China put forward for the first time in the "Decision of the Central Committee of the Communist Party of China on Comprehensively Deepening Major Issues of Reform": "Take grid management and socialized services as the direction, improve the grassroots comprehensive service management platform, and timely reflect and coordinate the interests of the people at all levels". Grid management is a major practical innovation of grassroots social governance in China since the 18th National Congress of the Communist Party of China. The 19th National Congress of the Communist Party of China and the Third Plenary Session of the 19th Central Committee further put forward the statement to "integrate related functions to set up comprehensive institutions and implement flat and grid management". On the basis of gradually building a complete grid governance system, the 20th National Congress of the Communist Party of China emphasized that in order to improve the social governance system and enhance the efficiency of social governance, it is necessary to "improve the grassroots governance platform supported by grid management, refined services and informatization".

How do we realize the high efficiency, refinement, and intelligence of social management? The "grid" is the basic management unit defined by the informatization of urban municipal supervision, and its purpose is to realize the fine management of regional blocks. Grid management divides urban space and urban management scope into grids and uses digital technology to process complex social governance affairs and social facts, so as to improve the clarity of urban grassroots social governance and make the refined management of small areas become a reality. Community grid management, as a grassroots social governance model, can better improve the social governance mode and innovate

the social governance system, and then can be widely implemented in major cities across the country.

One of the core components of grid management is the scientific division of grid cells. Currently, the 10,000 m unit grid division method is mainly used in various places, but this method can roughly distinguish the differences among grids. However, the differences in the built age, internal structure, geographical position, and surrounding environment of communities create obvious differences between grids and cause substantial differences in the difficulty and workload of management. In order to implement grid management units and enhance grid management capabilities, it is necessary to explore the construction of an assessment system serving grid governance, and further distinguish the differences of different grid units on the basis of the existing 10,000 m unit grid division. The quantitative assessment function of grid units is helpful to optimize grassroots public service management and promote the modernization of grassroots governance. Meanwhile, with the stable development of our society, we should meet the people's needs for high-quality life and spirit, create a stable, safe, and healthy community living environment through refined governance, and improve the happiness index of residents.

## 2. Literature Review

Scholars have had a lot of valuable discussions on the advantages and practical difficulties of grid governance, the connotation of refined management, and the realization path of refined management.

With regard to the advantages and practical dilemmas of grid governance, based on modern information technology as the supporting point, grid governance divides urban communities into small unit grids, so as to realize fine management through the subsidence of administrative power. In comparison to traditional grid management, it is more refined with respect to management units, responsibility and authority, management informatization, information visualization, management and service precision, and management tool standardization [1]. However, as the governance cost of grid governance informatization increases with the growth of information amount, basic information technologies such as electronic maps and mobile phone positioning must be fully utilized for support. Therefore, grid governance has limitations in terms of information governance costs. Based on the complexity of community events and the motivation and capability of individuals and organizations to conceal or distort information, grid governance involves certain limits regarding the informatization of social facts and the uncertainty of information utilization [2,3].

In the respect of the connotation of fine management, the Shanghai Grid has proposed the achievement of multiple forms of co-governance from three levels such as concept, information, and main body, and on that basis, implement fine governance to realize block coordination, coordination between government and community, and information coordination, thereby promoting the transformation of government functions, the innovation of the social system and mechanism, and the sharing of big data technology, and making up for the shortcomings of social development [4]. Secondly, grid management makes up for the deficiency of government management functions and is capable of promoting social repair. Bureaucratic fine governance is an important measure to reform "Guan Jin Min Tui" and make up for the lack of community repair ability [5]. Community grid governance, in the prevention and control of the COVID-19 outbreak, makes use of administrative means, legal means, persuasion and guidance means, and technical means to significantly improve the breadth, speed, and accuracy of policy implementation coverage, and improve the prevention and control capacity of communities through means of fine governance [6].

When it comes to the realization path of the community grid, a quantitative analysis on the effect of community grid governance was carried out in Jiao District, Jiamusi City, in which it was proposed that it was necessary to build an intelligent information platform to enable grid governance, and strengthen institutional protection and build a high-level grassroots team [7]. Community grid governance can also be applied in three aspects such as platform construction, data construction, and key applications through GIS technology

to enable community grid data division and population and housing data registration in the community to realize population and housing data standardization. Moreover, with measures such as spatiotemporal data dynamics, it can provide reference for the work supervision of grid workers, the division of communities, the allocation of educational resources, and the feedback of regional problems [8,9].

According to the viewpoints of all parties, domestic scholars mainly discuss the advantages and practical difficulties of grid governance, the connotation of refined management, and the realization path of refined management, while foreign scholars have less related research, mainly studying the formation of communities and governance models. With the continuous development of increasingly rich material life, people's requirements for life are gradually improving. Therefore, how to further improve the level of community grid governance has become a difficult problem of grid governance refinement. Grid governance cannot be refined further. The main reason is that there are differences in residents' income, community area, number of houses, and other indicators in the community, which lead to different collection quantities of community residents' information and service items to assist residents in the governance process, thus affecting the number of grid members dispatched and the daily workload of grid members.

## 3. Materials

### 3.1. Governance Model

The grid has realized the sinking of the focus of government services, properly solved the details that the government cannot handle, and at the same time, gave full play to the characteristics of grid management, quickly found problems, and finely handled events by virtue of the grid staff's high familiarity with the community and the accurate analysis of the information platform, and efficiently handled various problems. The exploration of grid management in China began in 2004, and a pilot project of community grid management was established in the Dongcheng District of Beijing. On this basis, the establishment and acceptance of pilot projects have been carried out nationwide in three batches, among which there are four most representative grid management modes, namely, digital development mode—represented by the Dongcheng District of Beijing, party building leading development mode—represented by Shanghai, group service development—represented by Zhoushan of Zhejiang, and refined development mode—represented by Yichang of Hubei [10–14].

#### 3.1.1. Digital Development Model

Beijing has established a management mode of "four-level management and three-level platform" by means of grid information, namely, a three-level information support platform of a district-level integrated command center, street sub-center, and community workstation, so as to realize grid service management by users at district, street, community, and grid levels. In terms of information construction, the Dongcheng District has established a trinity social service management information support system of a cloud computing center, grid geocoding, and Internet, so as to achieve vertical coverage to the end and horizontal coverage to the edge. In the aspect of management, by integrating government functions, an urban management supervision center and an urban comprehensive management committee are established from two aspects of supervision and management, and the Dongcheng District is managed with two "axes" to ensure that the supervision system and the management system are independent of each other and perform their respective duties. In terms of working procedures, the Dongcheng District of Beijing put forward the guiding idea of grid-based social service management of "refined management, humanized service, standardized operation and information support", following the five guiding principles of standardization, intensification, synergy, service, and visualization. At the same time, a six-step closed-loop business collaboration method is proposed, namely, six-step closed-loop business processing, which comprises discovery and reporting, command and dispatch, disposal feedback, task verification, assessment, and statement filing. Finally, the Dongcheng District has established a scientific and perfect supervision and as-

sessment system to comprehensively evaluate all aspects of community management. The digital development model of community grid management represented by the Dongcheng District of Beijing is shown in Figure 1.

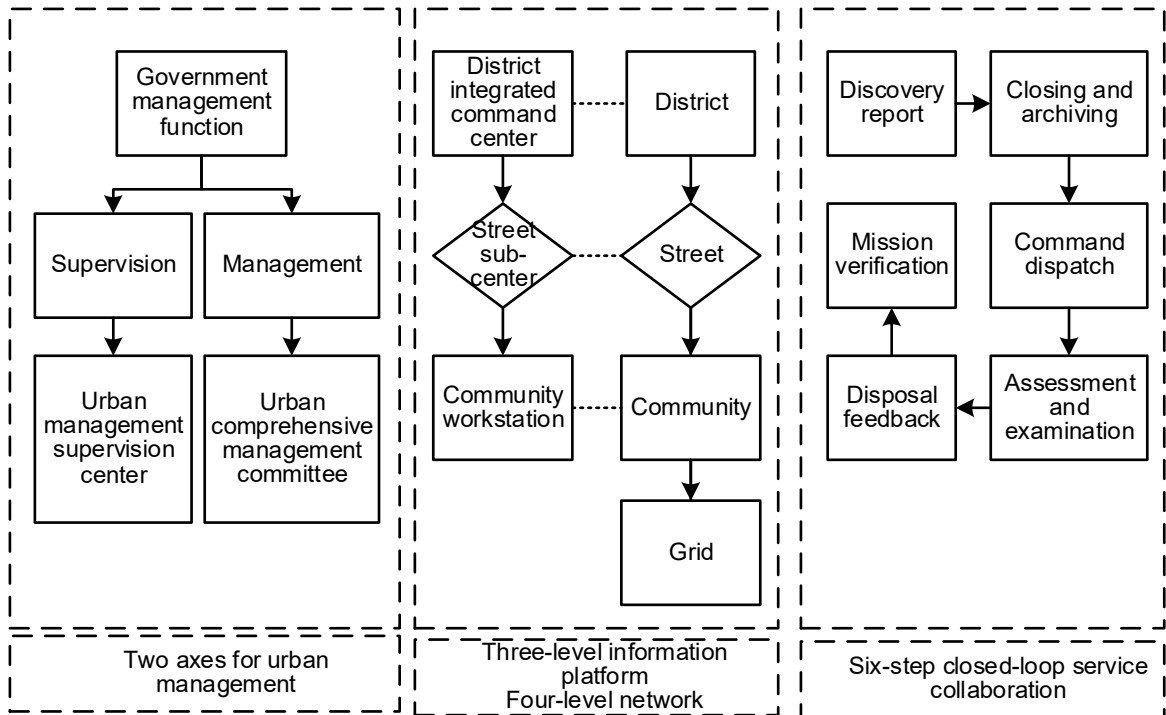

**Figure 1.** Community grid management mode diagram of Dongcheng District, Beijing.

### 3.1.2. Party Building Leads the Development Model

Shanghai's community grid management implements the management mode of "two levels of government, three levels of management and four levels of network". On the basis of retaining the experience of the community grid in Beijing, we should optimize the third-level management, namely, strengthen the leading core position of street party working committees, expand the practical space for party organizations to participate in grassroots communities, and enhance the influence of the party in the community more comprehensively. At the same time, the Street Party Working Committee was transformed into the Community Party Working Committee, which served as the core leadership force of community grid management and carried out full coverage management in three aspects: organization, work, and system. At the same time, the Changning District innovates and implements fourth-level management, constantly improves the community governance structure, and makes the power of political parties and various resources form more effective integration under the carrier of the "grid". Absorbing Beijing's advanced experience in system construction, the management system of "two axes" and "one platform" has been established. The Changning community involves 13 functional departments, including the police station, office of industry and commerce, office of environmental sanitation, urban management detachment, real estate office, and traffic patrol police, and implements the "6 + 7" "gradual grid" mode, and established a command center and emergency call system to ensure the safety and happiness of residents. At the level of rules and regulations, the articles of association of residents' autonomy are improved, and the nature and functions of neighborhood committees are defined. At the same time, various work systems such as operation manuals, management standards, and job responsibilities are formulated for grid management, so as to manage the implementation of grid members more standardly and efficiently. The leading mode of the community grid management party building represented by the Changning District of Shanghai is shown in Figure 2, each dotted box represents a processing level.

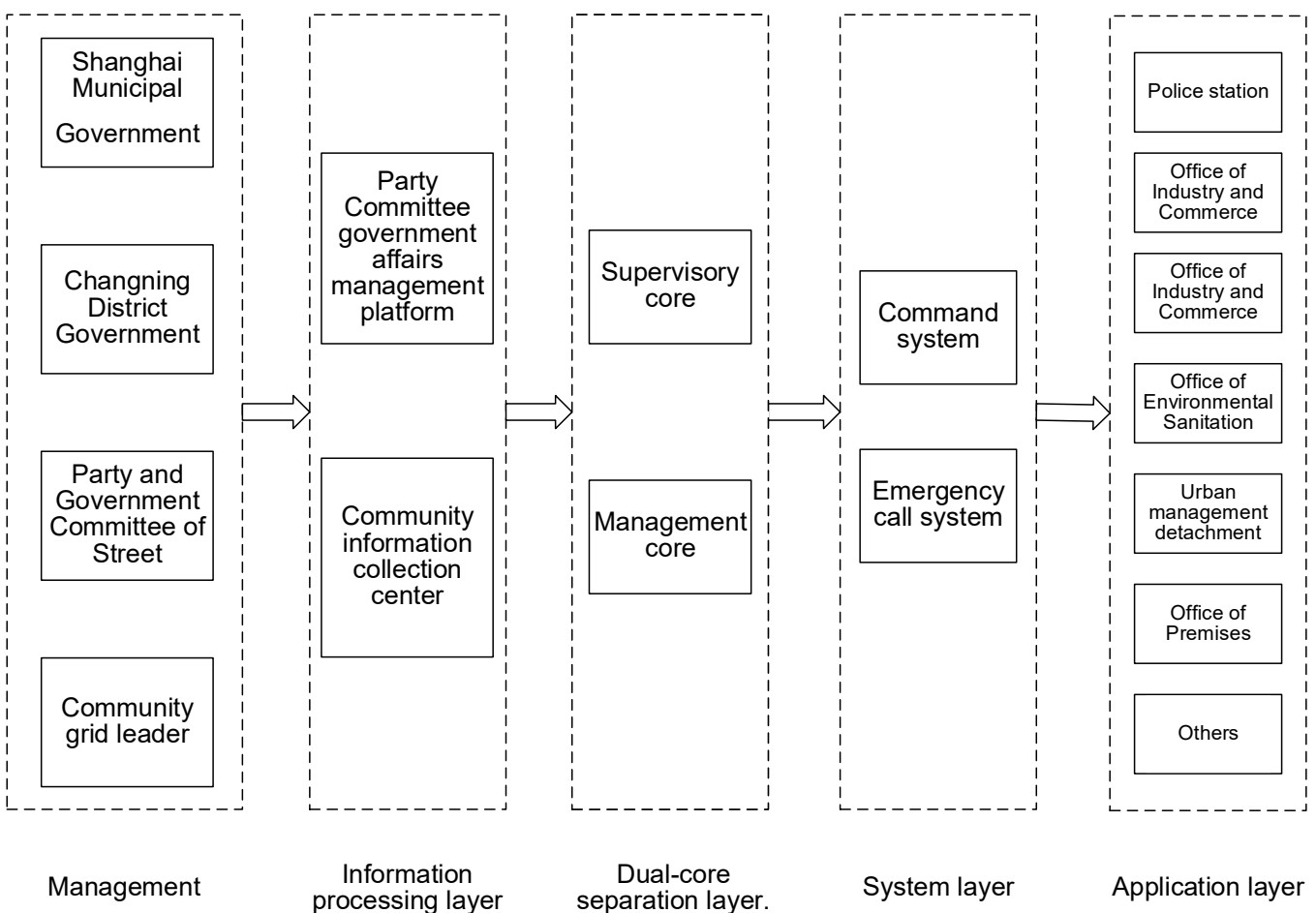

| Management | Information processing layer | Dual-core separation layer. | System layer | Application layer |

**Figure 2.** Community grid management mode diagram of Changning District, Shanghai.

### 3.1.3. Group Service Mode

The biggest feature of community grid design in Zhoushan City, Zhejiang Province is the introduction of group service into the original grid management, namely, taking people as the center and establishing a shared, diversified, information-based, omni-directional, and normalized community governance model. Zhoushan has established a five-level network and a "work leading group" at the municipal level, headed by the secretary of the municipal party committee or the mayor. The leading group has an office leading five special groups, namely, the comprehensive management and safety group, team management group, urban working group, fishing and rural working group, and technical support group, which are, respectively, under the responsibility of the Political and Legal Committee, the Organization Department of the Municipal Party Committee, the Civil Administration Bureau, the Municipal Agriculture and Fisheries Office, and the Municipal Information Center. Each grid is equipped with a service team consisting of 6–8 people to maximize the integration of resources and respond to the demands of the masses in a timely manner. At the level of information technology, Zhoushan City has developed a comprehensive network platform, which has the functions of data query and statistics, information interactions, work exchanges, and online services. The platform is divided into modules such as "basic data", "service", "short message interaction", "work exchange", and "public sentiment log", which enables the municipal government to dynamically control the actual situation of the community, and also enables residents to cooperate with one network and handle business at one time. In addition, residents' feedback to the community will be uploaded to the platform in time by means of SMS, telephone, and grid staff visits, so as to realize two-way supervision and two-way assessment of community governance, promote information communication, and improve the level of refined community governance.

The group development mode of community grid management in Zhoushan, Zhejiang Province is shown in Figure 3, the rectangle represents the actual institution, organization, and system, and the diamond with dotted lines represents the characteristics and functions of the model.

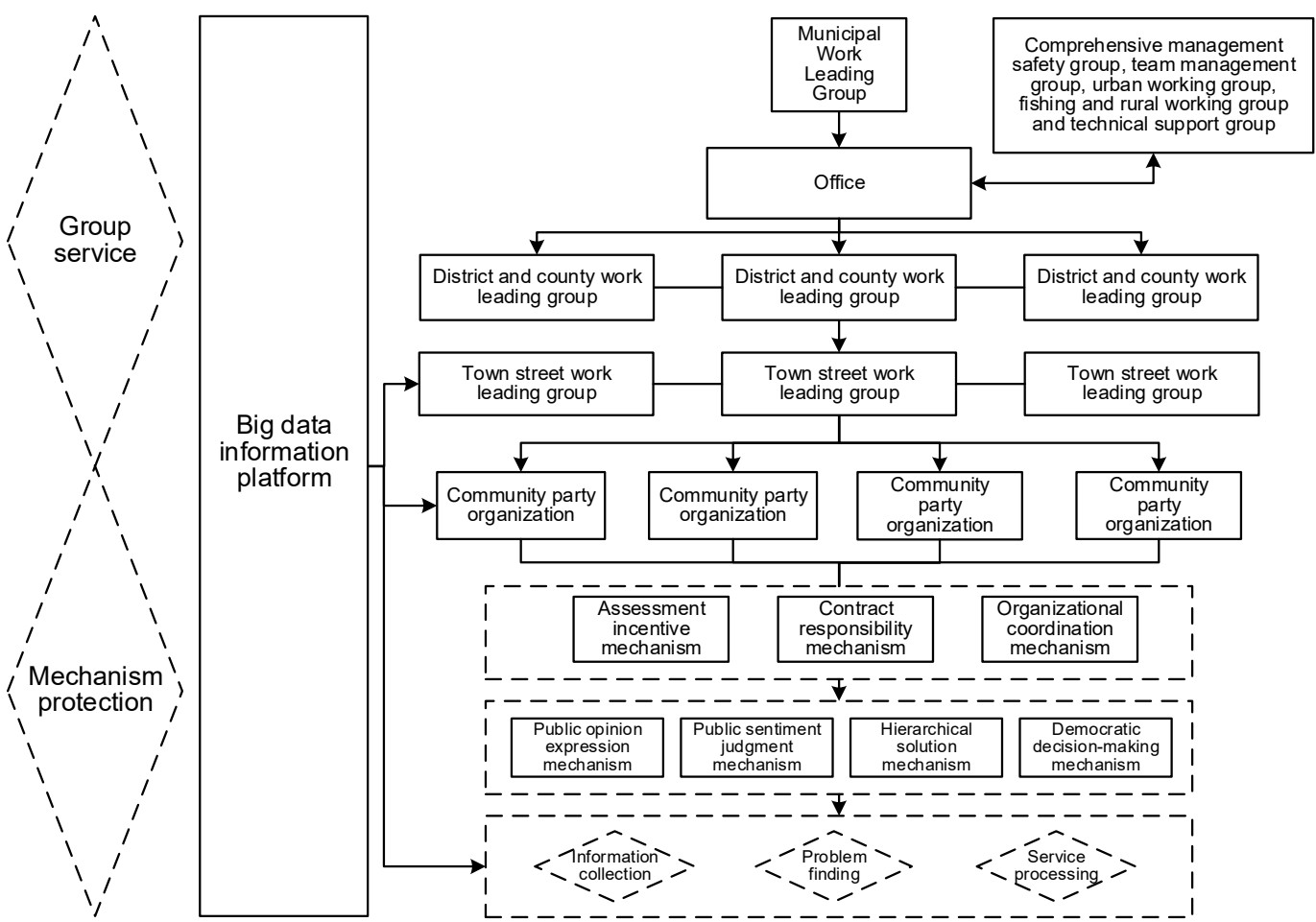

**Figure 3.** Development model of Zhoushan Community Grid Management Group in Zhejiang Province.

### 3.1.4. Refined Development Model

Yichang, Hubei Province has set up a "community comprehensive information service management platform" with four-level networking of "city–district–street–community". At the same time, it has set up "three centers" at the street level, including a people's service center, a comprehensive petition and stability maintenance center, and a grid management center, and established "three teams" at the community level, namely, community full-time workers, grid administrators, and community volunteers. In terms of the management system, its basic principles can be summarized as "one book, three transformations": "one book" means adhering to "people-oriented" practices, and "three transformations" means adhering to "grid management", "information support", and "whole-process service". In terms of human resources, Yichang recruits grid administrators through the public according to the principles of openness, fairness, voluntariness, and merit. In addition, the Yichang government has specially formulated the Training Program for Recruitment of Community Grid Administrators in Yichang City, which regulates the work requirements of grid members in examination and employment, exchange training, job responsibilities, discipline requirements, wages and salaries, assessment rewards and punishments, staffing, social security, termination and renewal of contracts, etc. Finally, they have established a professional mediation talent pool, further standardized the employment conditions, remuneration calculation, and assessment criteria, and provided talent reserves and conve-

nient conditions for professional and technical personnel in various fields to participate in mediation. At the level of community service, Yichang has established an access mechanism for community public services and a government purchase mechanism for services, has purchased community services through financial payment, has introduced high-quality services into the community, and has realized the continuous improvement of community service quality. The fine development mode of community grid management in Yichang, Hubei Province is shown in Figure 4.

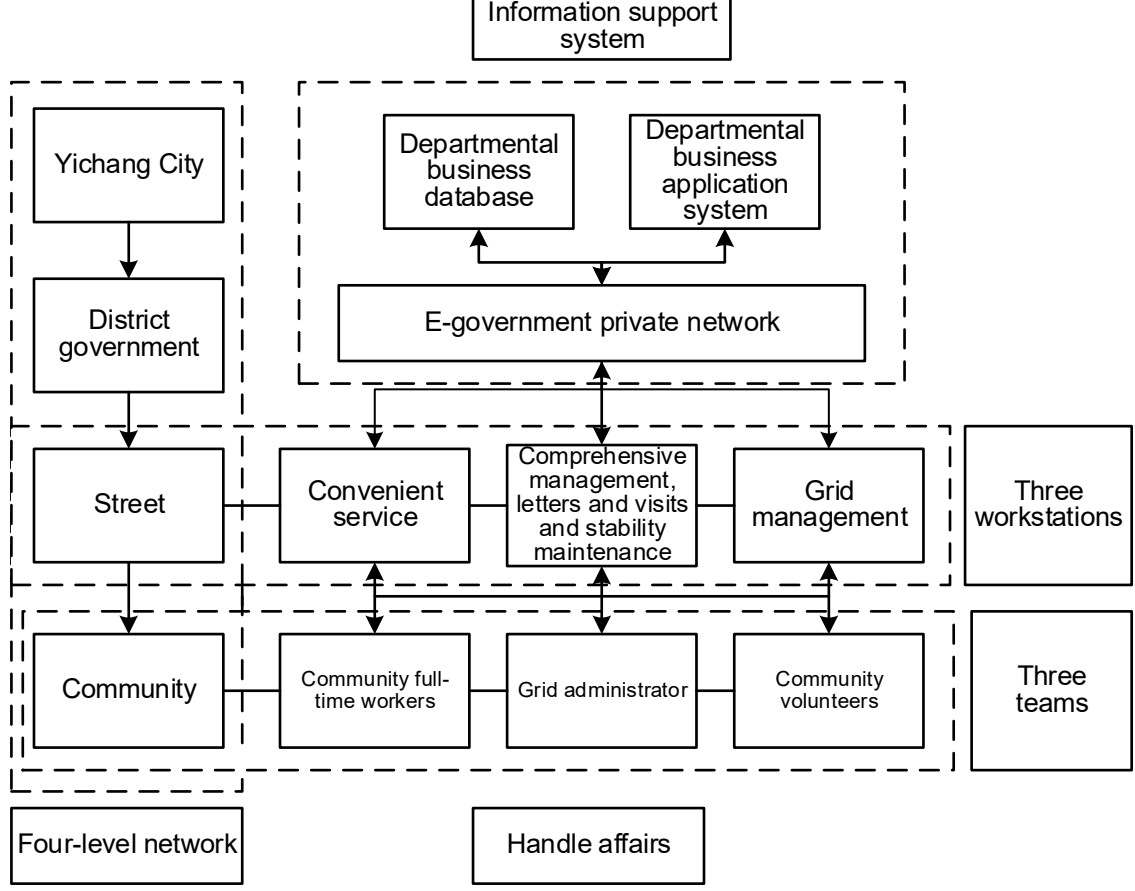

**Figure 4.** Grid governance model of Yichang community in Hubei Province.

*3.2. Governance Tasks*

The refined governance of the community grid has become an inevitable outcome of the times. The daily work of community grid members ensures the normal operation of the community and brings great convenience to residents' lives. Different from top-down traditional governance, grid refined governance sinks the focus of governance, and multiple departments work together and cooperate with each other to realize diversified professional governance based on big data. Reasonable division of labor among different departments and tasks in the community can better clarify the responsibilities among departments and reduce the communication errors between departments. At the same time, the assessment of community tasks can assign different grid members to different communities and realize personalized management for a single community. Taking the daily work of grid members combined with the actual needs of residents as the standard, scoring according to the difficulty and urgency of different work is undertaken, so as to obtain the demand status of the whole community for grid members. Among them, the work and actual situation of grid members are taken as indicators, even if five indicators are found, such as serving the people, risk reporting, conflict persuasion, grid patrol, and problem investigation, and the secondary indicators composed of specific elements are

taken as the next assessment criteria, and each community is evaluated and scored to form a refined governance distribution scheme.

The policy changes in grid management in China's urban communities are roughly divided into three stages: incipient exploration (2005–2012), deepening and expansion (2013–2016), and precise optimization (2017–present) [15]. This paper retrieves relevant policy texts on community grid management from the Chinese government website, government portals at all levels, Puklaw, and CNKI, etc. With keywords such as "community grid management", "grid management program", "community workers' performance management methods", "grid supervisor", and "grid service management regulations, NVivo11 was used to analyze the central and local policy documents on community grid management and sort out the work content and work direction of community construction. Based on the word cloud analysis results shown in Figure 5, the current urban community grid governance is mainly developed in the direction of comprehensive governance, intelligent community, and service-oriented community, and the community is finely managed by an intelligent governance platform.

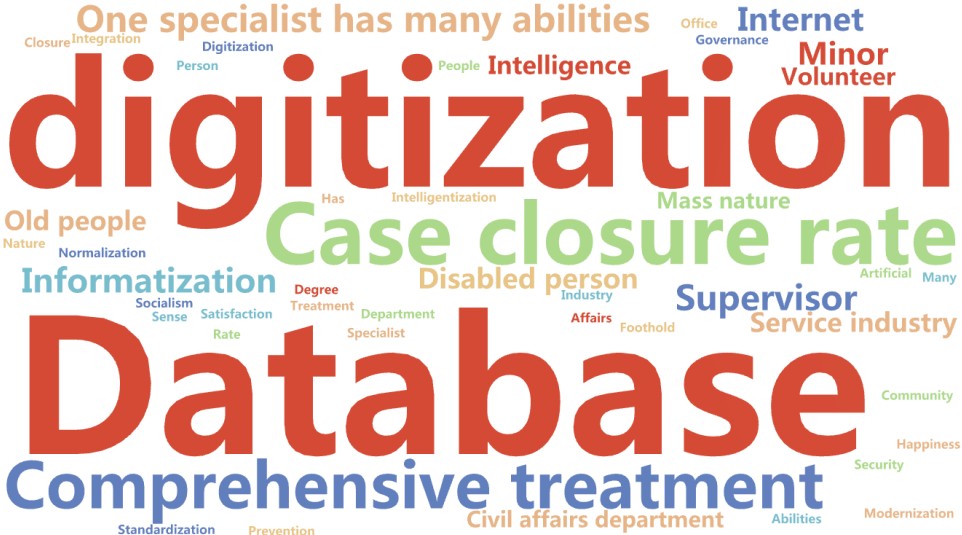

**Figure 5.** Word cloud of main tasks of community grid members.

This paper collected 134 urban community grid governance policy texts from the central to local levels published between 2013 and 2022. Twenty-six initial concepts of community grid management influencing factors were extracted from the policy texts. Through repeated induction and integration, the 26 initial concepts were summarized into 11 categories, which are the main factors affecting community grid governance. For example, the "comprehensive collection of community grid... feedback work" in the Management Measures for Urban Community Gridders in Anhui Province (for trial implementation) belongs to the information collection community governance elements, and the content analysis unit code of urban community grid governance policy text is shown in Table 1. The governance indicators of the urban community grid are shown in Table 2.

As for the 11 open categories, we found the relationship between them according to the policy text, and further refined and summarized these relationships into the main categories. By analyzing the correlations between the first-level codes and studying the logical relationships among them, the six main genera of "handy services for the public", "risk report", "conflict mediation", "grid patrol", "problem detection", and "community structure" can be extracted from the genera by combining the management tasks of community grid members.

Through the analysis of the main genus, five core elements were integrated: "handy services for the public", "risk report", "conflict persuasion", "grid patrol", and "problem detection".

**Table 1.** Example of content analysis unit code for urban community grid governance policy text.

| No. | Policy Text Name | Content Analysis Unit | Influencing Factors |
|---|---|---|---|
| 1 | Notice of Anhui Provincial Department of Civil Affairs on the issuance of Anhui Urban Community Grid Management Measures (for Trial Implementation) | Comprehensive collection of information on "people, places, things, objects and organizations" in the community grid, timely investigation and reporting of the urban environment, residents' demands, conflicts, and disputes, potential accidents and all kinds of emergencies in the grid, and good follow-up services and verification feedback | Information Collection |
| . . . | . . . | . . . | . . . |
| 3 | Notice of the General Office of Fuzhou Municipal People's Government on the issuance of Fuzhou City Grid Inspection Management Measures (for trial implementation) | Assist in visiting and investigating key groups such as correction offenders in the community, emancipists, drug addicts, and people with severe mental disorders | Handy services for the public |
| . . . | . . . | . . . | . . . |
| 57 | Notice of the Office of the People's Government of Haicheng District, Beihai City on the issuance of the implementation plan of the grid supervision of air pollution prevention and control in Haicheng District, Beihai City | Motivate residents, enterprises, and institutions in the area to create a good atmosphere for everyone to pay attention to the prevention and control of air pollution and participate in the prevention and control of air pollution, so as to effectively improve the prevention and control of air pollution | Community environment |
| . . . | . . . | . . . | . . . |

**Table 2.** Governance indicators of urban community gridding.

| No. | Grade 3 Code | Grade 2 Code | Grade 1 Code |
|---|---|---|---|
| 1 | Handy services for the public | Resident satisfaction | Regular petitions |
| | | | Handy services for the public |
| | | Intelligent building | Platform construction |
| | | | Information collection |
| | | | Facility construction |
| 2 | Risk report | Community safety | Fire safety |
| | | | Assistance in law enforcement |
| | | | Daily patrol |
| 3 | Conflict mediation | Policy promotion | Legal promotion |
| | | | Environmental promotion |
| | | | Fire control promotion |
| | | Organizational cultivation | Cultural activities |
| | | | Joint construction |
| | | Psychological construction | Pet management |
| | | | Neighborhood conflicts |
| | | | Thought leadership |

**Table 2.** *Cont.*

| No. | Grade 3 Code | Grade 2 Code | Grade 1 Code |
|---|---|---|---|
| 4 | Grid patrol | Community enterprise environmental protection | Community environment |
| | | | Corporate emissions |
| | | Community business operation | Violation investigation and punishment |
| | | | Impact on order |
| 5 | Problem detection | Community facility maintenance | Facility safety |
| | | | Illegal construction of facilities |
| | | Community operation | Municipal administration |
| | | | Fire engine access |

### 3.2.1. Convenience Service Representation

Service for the convenience of the people and the benefit of the people refers to problem assistance and trouble solving for the difficulties encountered by community residents in their daily life. There are many and complicated affairs in this area, and most of the events are very small, such as "inconvenience for residents to travel due to sewer blockage", "accumulation of sundries occupies fire exits", "elevator operation failure makes the elderly unable to travel", and other daily trivial matters. Classified according to types, these trivial matters can be divided into "health and environmental management", "chaotic parking", "chaotic selling and occupation of roads", "property maintenance", "special group services and family planning services", "neighborhood, lease, family, and neighbor disputes", "pet management", and "other events". The smaller the number of difficult groups and special groups in the community, the less daily details that need to be dealt with, and the number of grid members and the work intensity of grid members needed by the community will be reduced accordingly. Due to the different severity and urgency of daily chores, the number of chores in each community is also different. Therefore, with the background of creating a safe community, classifying and evaluating the community population can undertake a more targeted job of facilitating the people and benefiting the people, provide extra attention and labor for the elderly and families in difficulty, and ensure the daily life quality of community residents.

### 3.2.2. Characterization of Risk Report

The risk hazard report refers to the grid staff meeting with community safety personnel in daily life, and conducting safety risk investigation actions on houses, fire exits, and stores along the street in the community. Grid members are very familiar with community residents, basic structures, and the overall situation because they have been stationed in the community for a long time and can accurately understand community information and efficiently deal with various potential risks. The objects of inspection mainly include self-built houses in the jurisdiction, potential safety hazards of electricity consumption in communities, use of fire-fighting equipment, smoothness of escape emergency exits, potential safety hazards of gas facilities, number of elevators, etc. The number of these infrastructures and the construction years of community infrastructure also vary from community to community. The larger the overall size of the community and the more infrastructure, the greater the workload of grid members. If the community belongs to an old community, it is necessary to pay more attention to the safety of facilities to avoid the safety problems caused by the aging of equipment.

### 3.2.3. Characterization of Contradiction Persuasion

General contradiction persuasion refers to the mediation of disputes arising from neighbors, families, properties, etc., by grid members so as to reduce the problems caused by friction between people and avoid the escalation of small things into intractable major

events. These kinds of problems mainly appear in the disputes between residents and public spaces in the community, and so the number of houses, the length of roads, and the area of underground parking lots in the community are very important indicators. Through these indicators, the size and population density of the whole community can be clearly estimated. The smaller the number of the two, the less likely the community is to have conflicts, the smaller the number of grid members to be sent, and grid members need to undertake less difficult tasks. Therefore, taking the general contradictions in the community as the starting point, evaluating the number and scale of the community can effectively carry out refined governance of the community, so that the contradictions in residents' lives can be mediated in time.

### 3.2.4. Grid Patrol Representation

Grid normalization patrol means that grid members regularly inspect their jurisdictions, mainly to prepare for prevention and crackdown, to discover and report illegal activities, various street crimes, and mass incidents in time, and to remind residents to close doors and windows, and monitor water, electricity, and gas during patrols, so as to improve safety awareness and achieve full coverage of social services. The important indicator of grid inspection in the community is mainly the number of houses. At the same time, attention should be paid to the special organizations in the whole community and surrounding jurisdictions, namely, compulsory education schools, training institutions, medical institutions, markets, fitness places, and operating stores. This kind of place has a large flow of people and miscellaneous types of people, which is a key place to pay attention to and a high-risk place where incidents occur. Therefore, problems encountered in such places need special attention. After defining the indicators such as community size and number of special organizations, we can clearly evaluate the community, send a reasonable number of grid members, conduct daily inspections in conjunction with police stations, and pay extra attention to key working places and key places to ensure the safety and stability of the community.

### 3.2.5. Characterization of Problem Investigation

Problem investigation and discovery refers to the task that grid members carry out in finding hidden dangers in communities and jurisdictions to ensure residents' life safety and improve residents' life satisfaction. Problem investigation mainly includes cleaning up advertisements, ensuring road safety, checking fire-fighting facilities, checking dangerous electricity consumption, cleaning up accumulated combustible materials, etc., such as "small advertisements are posted indiscriminately on walls, which affects the appearance of the city", "missing road manhole covers easily lead to residents falling down", "roadside bike-sharing stops indiscriminately blocking fire exits". Public places around the community are often the hardest hit areas where problems occur. Therefore, taking the number of bus stations, subway exits, parking spaces on the ground, refueling stations, green areas and underground parking spaces as indicators can effectively evaluate the number of grid members needed by the community in problem investigation and discovery, and send grid members to investigate problems in time to solve potential safety hazards.

### 4. Methods

Community grid members need to deal with complicated and detailed matters, but there are differences in population, age, poverty, education level, infrastructure area and quantity, old and complex community structures, etc., which leads to low work efficiency and insufficient staff in the daily work of grid members, which affects the operation efficiency of grid management. Community grid unit assessment mainly refers to dividing communities according to grids at the level of community governance, and then rating communities on this basis. Fine management refers to a management mode that minimizes the resources occupied by management and takes reducing management costs as its main management goal. Community units take the grid as a bridge to formulate personalized

management schemes for each community and realize digital control and diversified linkage governance with the help of a network. At the same time, according to the basic situation of the community, the number of grid members dispatched and the division of tasks are analyzed, which can effectively alleviate the governance dilemma, realize refined governance, and improve the grid operation efficiency.

The management personnel of the refined community grid are classified based on the severity of events, namely, "general emergency", "ordinary event", and "major event", and the urgency of events, namely, "general emergency", "non-emergency", and "emergency" [5]. At the same time, considering the workload and work intensity of grid members, the daily work of grid members is classified, and the communities with greater difficulty are screened to send out additional staff, and the allocation of personnel is also increased for tasks with high urgency. At the same time, relying on grid characteristics, information technology is applied to ensure multiple departments actively cooperate and human resources are reasonably used in appropriate places, minimizing the problems of grid members' repetitive work and low work efficiency, and minimizing residents' difficulties in doing things, especially for those who are unwilling to rely on the grid, are distrustful of the grid, and are unfamiliar with grid.

### 4.1. The Level of Facilitating the People and Benefiting the People

Taking the service index for the convenience of the people and the benefit of the people as the first-level index, four second-level indicators were established, namely, demand collection and feedback, basic information of residents, special population service, and agency affairs for residents. At the same time, twelve categories, including residents, population, floating population, special groups, school-age children, elderly people, flexible employment, ethnic minorities, party members, United Front objects and retired soldiers, were taken as three-level indicators. Among them, the special population was subdivided into five categories: drug addicts, mentally retarded persons, released prisoners, correctional personnel, and key personnel, and the number of difficult groups was subdivided into six categories: difficult families, disabled persons, lonely persons, families who have lost their independence, left-behind women, and difficult children.

### 4.2. Risk Reporting Level

The potential risk reporting index was taken as the first-level index, and the infrastructure investigation was taken as the second-level index. The second-level indicators were concretized, and the third-level indicators were established, namely, the number of elevators, the number of communication base stations, gas facilities, power-on facilities, temporary storage places for discarded goods, fire exits, electric vehicle parking lots and public toilets.

### 4.3. Contradiction Persuasion Level

The general contradiction persuasion index was taken as the first-level index, and the mediation of contradictions and disputes was taken as the second-level index. Four three-level indicators were established: the number of houses, the length of roads, and the area of underground parking lots. Among them, the number of houses is subdivided into the number of houses with elevators and the number of houses without elevators.

### 4.4. Grid Patrol Level

We took the grid normalization inspection index as the first-level index and established three second-level indicators: enterprise investigation and reporting, publicity of policies and regulations, and community security and stability. The second-level indicators were subdivided into six third-level indicators: compulsory education schools, training institutions, medical institutions, markets, fitness places, and operating stores. Among them, compulsory education schools were subdivided into junior high schools, primary schools, and kindergartens, markets were subdivided into convenience markets, profes-

sional markets, and business districts, fitness places were divided into community squares and fitness centers, and operating stores were divided into restaurants, material return points, small workshops, gas stations, Internet cafes, and six other items.

### 4.5. Problem Investigation Level

We took problem investigation and discovery as the first-level index and established three second-level indicators: public facilities maintenance, sanitary environment maintenance, and property maintenance coordination. Then, the secondary indicators were subdivided, and the tertiary indicators were established, namely, the number of secondary water supplies, the number of gas users, the number of bus stations, the number of subway exits, the number of parking spaces on the ground, the number of electric vehicle parking lots, the number of refueling stations, the underground space area, the green area, and dozens of temporary waste stacking points.

Taking the daily work of grid members as the assessment index, the work content was divided into three levels of indicators, which were subdivided layer by layer. According to the corresponding elements in the three levels of indicators, the community was evaluated, and the level of the community was effectively allocated, namely, ABCD. This ensures the reasonable distribution of reasonable grid members to different communities, and fine treatment of community work. The index table of refined governance is shown in Table 1.

## 5. Discussion

The constructed assessment system of urban community grid units can not only differentiate the grid units but can also rationally allocate resources such as labor, materials, power, and financial resources according to the rating results, so as to realize scientific decision-making refinement, and also contribute to the development of grid refinement governance-related work.

### 5.1. Refinement of Grid Management Performance

Performance is the general name for benefit, efficiency, and effectiveness. Community governance is the application of governance theory at the community level, and its connotation is the process that the relevant stakeholders of the community complete and realize the management and service of community public affairs together around community public problems [16]. Community governance performance includes the governance process and governance results. As far as the community governance process is concerned, a performance assessment includes whether the number of grid members dispatched is scientific and reasonable. As far as the results of community governance are concerned, a performance assessment should pay attention to the number of residents' detailed problems that are solved, the efficiency of completion, and if the quality of completion meets the expectations of residents. The assessment system of the community grid unit can make clear the number and difficulty of details that grid members need to deal with in detail. According to the different community levels and the actual situation, it can distinguish emerging commercial housing communities, old commercial housing communities, affordable housing communities, and demolition and resettlement residential areas, and reasonably evaluate the difficulty of grid members [17]. With the help of a big data platform, we can track and accurately grasp the work processes of grid members in real time and give a clear calculation of the daily work performance of grid members. At the same time, community residents can upload their satisfaction to the big data platform, put forward opinions on grid management in time, and form community–residents two-way governance. The logic diagram of grid management performance refinement assessment is shown in Figure 6.

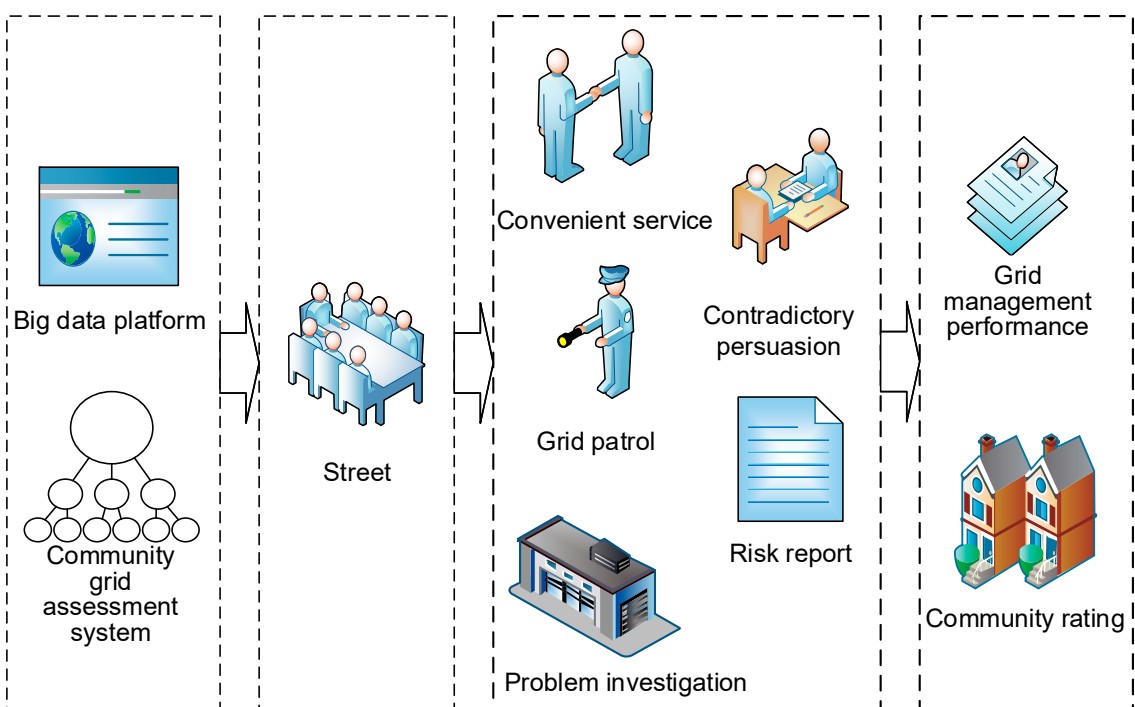

**Figure 6.** Logic diagram of refined assessment of grid management performance.

### 5.2. The Actual Needs of the People Are Accurate

The community grid assessment system can fully collect the actual needs of the people in the process of use, summarize and analyze them in time, provide refined services to community residents, and improve the level of community governance. As the smallest unit of social governance, community is one of the important factors in social stability, and the sense of security and happiness of community residents comes from solving pain points in the community. The "Opinions on Strengthening and Perfecting Urban and Rural Community Governance" pointed out: "All major decision-making matters involving the public interests of urban and rural communities, practical difficulties and contradictions and disputes related to the vital interests of residents, in principle, are led by community party organizations and grassroots mass autonomous organizations. Organize the residents to negotiate and solve them". It can be seen that residents are an important part of participation in community consultation, and it is an inevitable requirement of community grid governance to ask for the needs of the people and ask for the consideration of the people. Creating a community network assessment system can accurately evaluate the needs of the community and the people, and truly realize active governance and undertake it first. At the same time, the community grid assessment system combined with the big data platform can query the basic information of the number of community party members, low-income families, left-behind children, and families in difficulty in real time, and the people can also know the policy information in time with the help of the big data platform, so as to realize two-way communication and community public services combining both online and offline services. To sum up, the logic diagram of people's needs is shown in Figure 7.

### 5.3. Refinement of Enterprise Services

Realizing enterprise refined service refers to understanding the business status and existing problems of all enterprises within the jurisdiction with the help of the community grid and community grade assessment system, creating a high-quality business environment, effectively helping enterprises to improve efficiency of operations and management, and helping to formulate and adjust enterprise strategies. Firstly, the system can actively push relevant policies. The establishment of the community network assessment system can accurately locate the industry where the enterprise is located with the background

of understanding the basic situation of the enterprise, and the enterprise can fully know various preferential conditions such as incentive policies, subsidy policies, tax incentives, and employment subsidies, so as to achieve high-quality development of the enterprise. Secondly, the system can help enterprises to publicize their business through multiple channels. The system can rely on the Internet platform to push enterprise-related information to community residents, increase the opportunities for residents to get into close contact with enterprises, and further expand the influence of enterprises. Thirdly, according to the actual situation of the community, the community grid assessment system can analyze the enterprise services most needed by the community, help residents achieve a convenient life, and help enterprises seize development opportunities. Finally, the system can track the development of enterprises in the grid to help enterprises remove of difficulties. The community grid assessment system can also monitor the development of community enterprises in the grid in real time, help community grid members understand the development status of enterprises around the community, provide one-on-one assistance to enterprises, and effectively help enterprises solve practical problems and solve development problems. To sum up, the enterprise service logic diagram is shown in Figure 8.

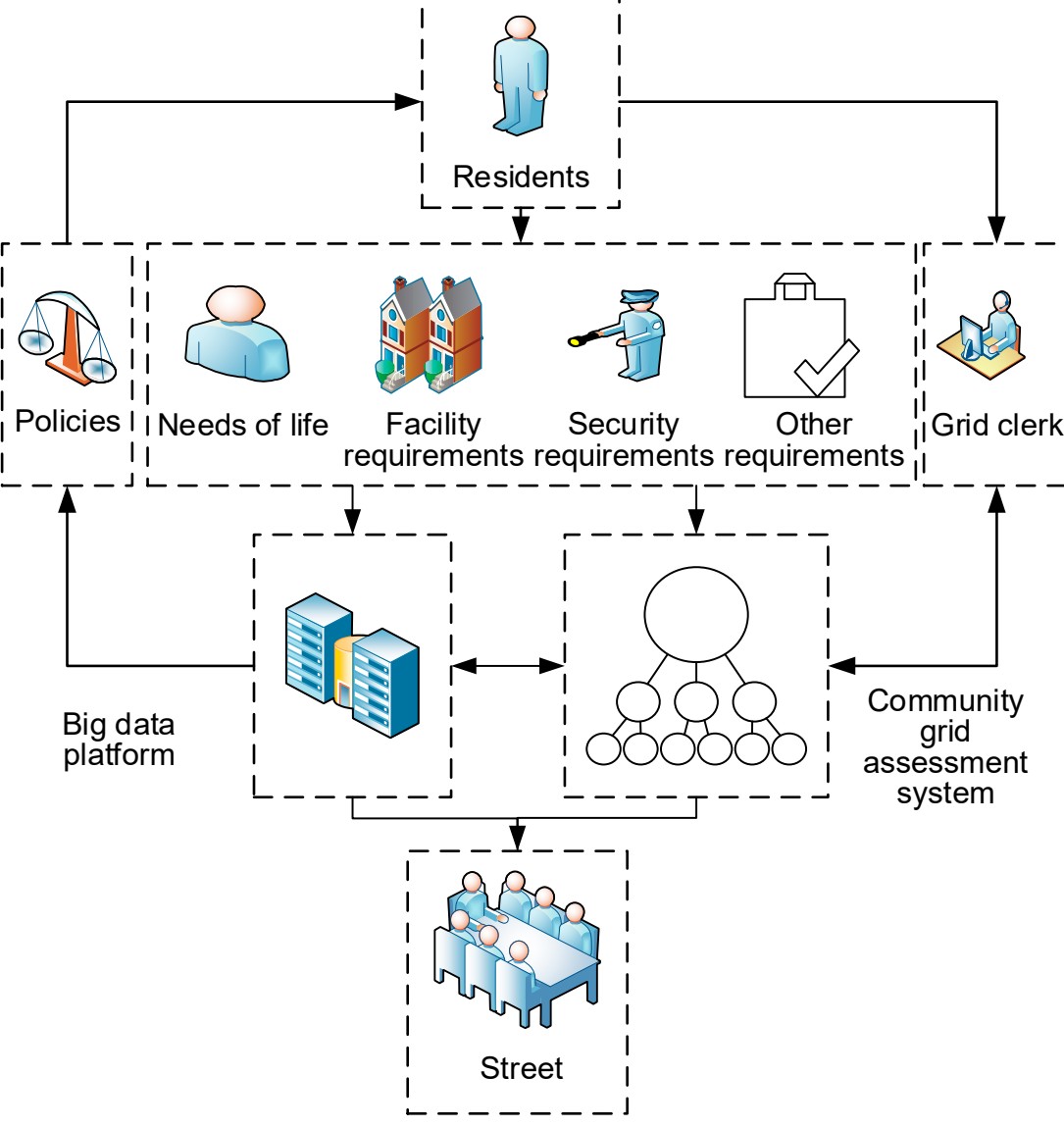

**Figure 7.** Logic diagram of people's needs.

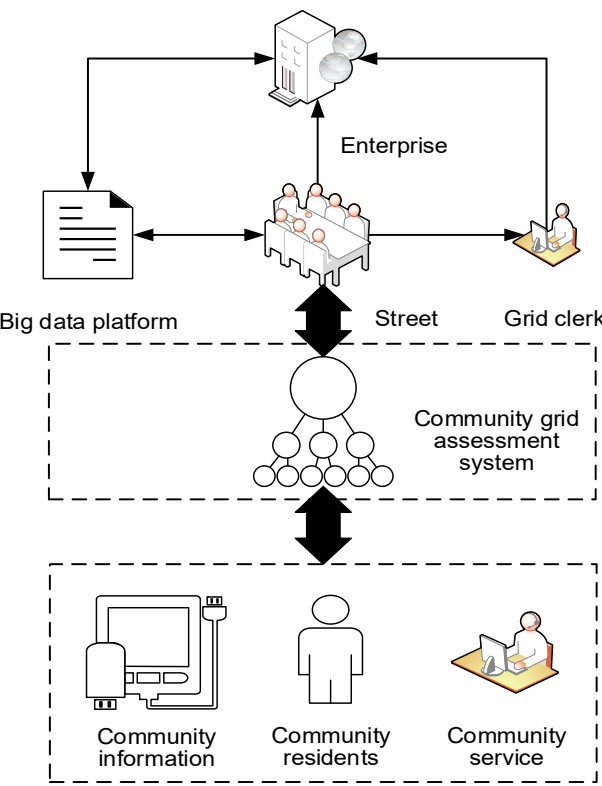

**Figure 8.** Enterprise service logic diagram.

### 5.4. Refinement of Non-Profit Organization Services

Service refinement of non-profit organizations refers to the classification of communities through the community grid system and solves the pain points in communities at fixed points. The service of community non-profit organizations should be integrated with people's longing for a better life, so that the service of community non-profit organizations can better adapt to the new challenges and requirements of community governance. First of all, the system can realize the positioning push of the community. The community grid assessment system can cooperate with the big data platform to rate different types of communities and push suitable communities to non-profit organizations and service organizations that can provide help. Secondly, the system can standardize the recruitment of community volunteers. According to the pain points of the community, the timely release of the service information of non-profit organizations and the standardization of the process of community voluntary organizations according to law can be undertaken. Finally, the services of non-profit organizations should be implemented. The "Opinions of the Central Committee of the Communist Party of China and the State Council on Strengthening and Perfecting Urban and Rural Community Governance" puts forward: "Improve the supply capacity of community services". The establishment of the community grid assessment system can locate community pain points, avoid community neighborhood committees from unilaterally understanding community service as formalism, truly realize the service refinement of community non-profit organizations, and fundamentally improve residents' happiness. To sum up, the service logic diagram of non-profit organizations and the application general diagram of community grade assessment system is shown in Figures 9 and 10.

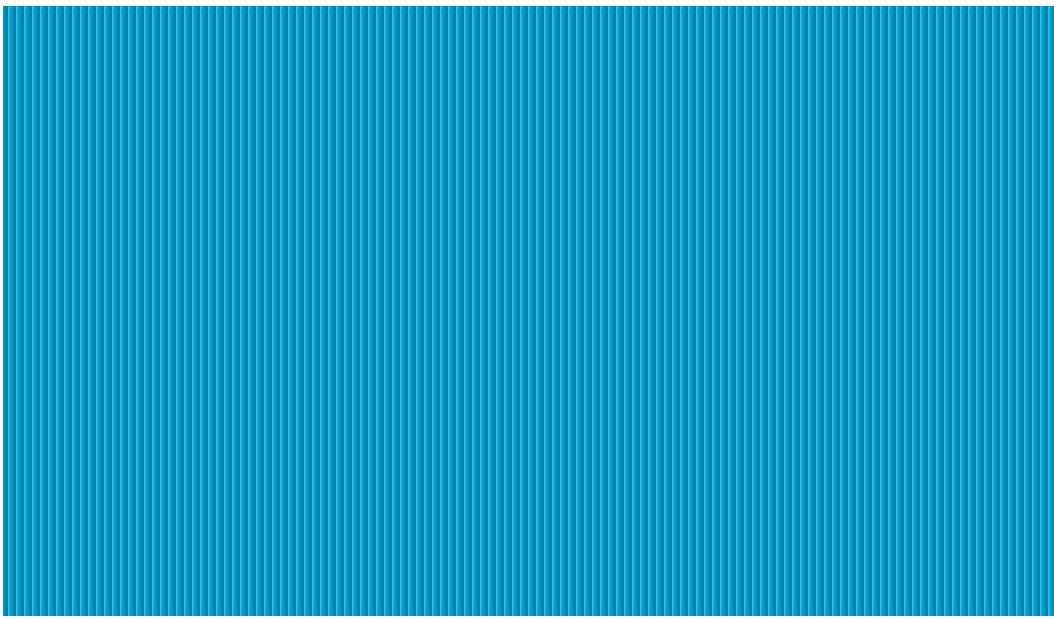

**Figure 9.** Logic diagram of voluntary service.

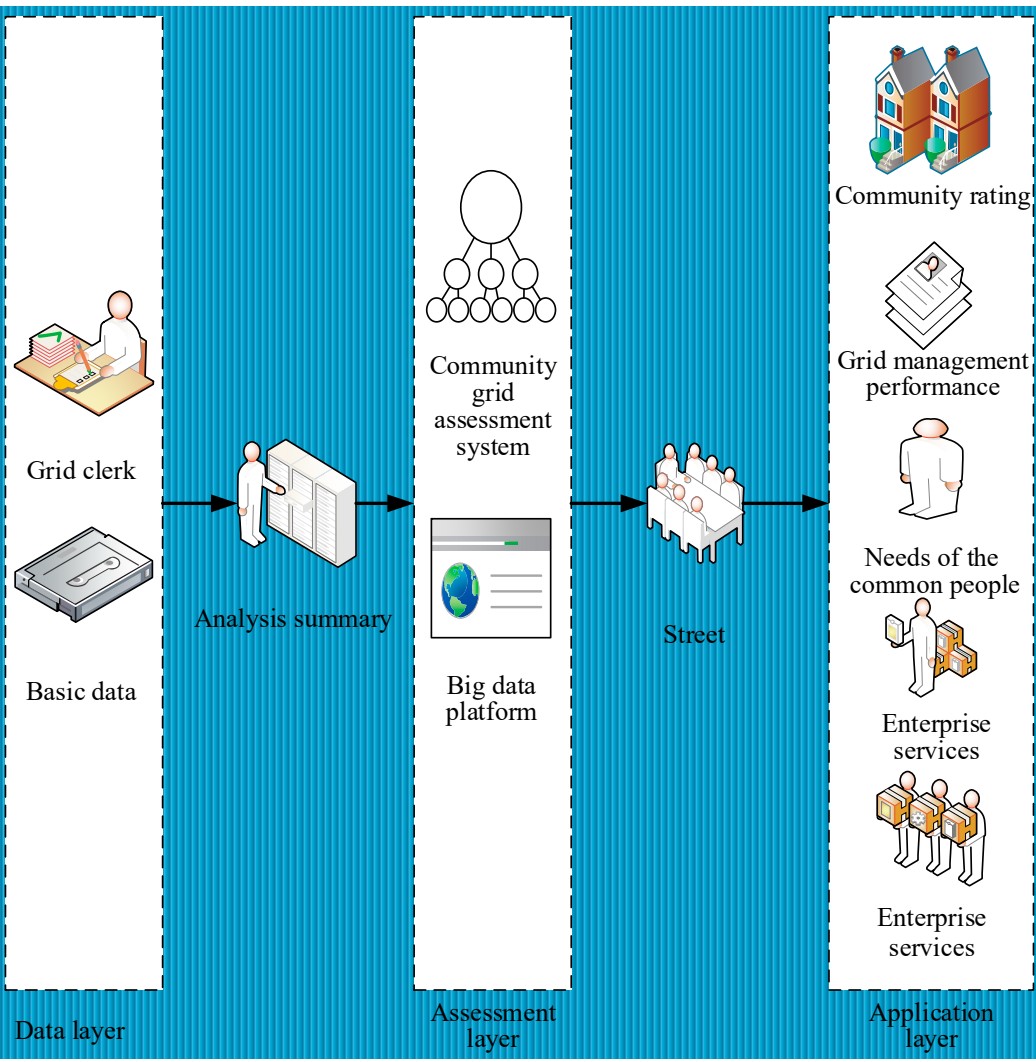

**Figure 10.** General drawing of application of community grade assessment system.

**Author Contributions:** Conceptualization, L.S.; methodology, L.S.; validation, L.S.; formal analysis, N.L.; investigation, N.L.; data curation, L.S.; writing—original draft preparation, N.L.; writing—review and editing, N.L. and L.S.; supervision, L.S. All authors have read and agreed to the published version of the manuscript.

**Funding:** This research was funded by Teaching and Research Project of China University of Geosciences (Wuhan), grant number 2013A42.

**Informed Consent Statement:** Informed consent was obtained from all subjects involved in the study.

**Data Availability Statement:** Not applicable.

**Conflicts of Interest:** The authors declare no conflict of interest.

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
