# Peer review of "Construction of Community Grid Unit Assessment System from the Perspective of Refined Governance"

_sustainability, doi:10.3390/su151310279_

Round 1

Reviewer 1 Report

Dear author,

From my point of view you should check your model, whit empirical dates  since it is more theorical with any empirical evidence and probably your conclusions will be better

As far as I know the English is fine, but from my point of view maybe check by native English

Author Response

Dear reviewer:

Thank you very much for your comments and professional advice. These opinions help to improve academic rigor of our article, Based' on your suggestion and request. we have made corrected modifications on the revised manuscript. We hope that our work can be improved again. Furthermore, we would like to show details as follows:

1.“Is the content succinctly described and contextualized with respect to previous and present theoretical background and empirical research (if applicable) on the topic? Can be improved”

Response: we have revised the sentence in the article. I have revised the literature review, supplemented the original sentences and contents, and added English references.

2.”Are all the cited references relevant to the research? Can be improved”

Response: It has been modified in the manuscript. The content of the references has been revised to make the article more appropriate to the references cited.

3.”For empirical research, are the results clearly presented? Not applicable”

Response: we have revised the sentence in the article. First of all, I supplement the source of data by constructing the index system in the article. Through NVivo 11.qualitative analysis software, based on Grounded Theory, the policy text is encoded at three levels. Secondly, because part of the data text comes from the government's undisclosed documents, which involves the confidentiality of the data, it is impossible to upload all the policy text data. Finally, this paper proposes an evaluation framework for community grid governance, which will be further supplemented in the future.

4.”Is the article adequately referenced?”

Response: It has been modified in the manuscript. English references have been added to the references section and text.

Thank you very much for your attention and time. Look forward to hearing from you.

Your sincerely,

Ningzhi Li,

School of Public Administration, China University of Geosciences, 388 Lumo Road, Wuhan, Hubei 430074, China

E-mail: [email protected];[email protected]

Reviewer 2 Report

It is a well written study.

Author Response

Dear reviewer:

Thank you very much for your comments and professional advice. These opinions help to improve academic rigor of our article, Based' on your suggestion and request. we have made corrected modifications on the revised manuscript. We hope that our work can be improved again. Furthermore, we would like to show details as follows:

1.”Are all the cited references relevant to the research? Can be improved”

Response: It has been modified in the manuscript. The content of the references has been revised to make the article more appropriate to the references cited.

2.”For empirical research, are the results clearly presented? Can be improved”

Response: we have revised the sentence in the article. First of all, I supplement the source of data by constructing the index system in the article. Through NVivo 11.qualitative analysis software, based on Grounded Theory, the policy text is encoded at three levels. Secondly, because part of the data text comes from the government's undisclosed documents, which involves the confidentiality of the data, it is impossible to upload all the policy text data. Finally, this paper proposes an evaluation framework for community grid governance, which will be further supplemented in the future.

Thank you very much for your attention and time. Look forward to hearing from you.

Your sincerely,

Ningzhi Li,

School of Public Administration, China University of Geosciences, 388 Lumo Road, Wuhan, Hubei 430074, China

E-mail: [email protected];[email protected]

Reviewer 3 Report

After reviewing the article, it is clear and precise how the methodology proposed is clearly delimited based on the objectives and hypotheses proposed by the authors. In addition, the empirical framework is current and clearly and precisely frames the scientific article. In this context, the authors' approach has allowed them to establish a theoretical model for the evaluation of three-level network units and the evaluation system of community network units. Moreover, the conclusions drawn on the basis of the research carried out are clear, allowing to solve problems of evaluation of the performance of community network members, among others. In this context I consider the article to be suitable for publication, although I would like the authors to mention in more detail and extensively the approach that would lead to future research on the basis of the results of the article.

Author Response

Dear reviewer:

Thank you very much for your comments and professional advice. These opinions help to improve academic rigor of our article, Based' on your suggestion and request. we have made corrected modifications on the revised manuscript. We hope that our work can be improved again. Furthermore, we would like to show details as follows:

1.”although I would like the authors to mention in more detail and extensively the approach that would lead to future research on the basis of the results of the article.”

Response: Thank you for your recognition of this manuscript. This paper proposes an evaluation framework for community grid governance, which will be further supplemented in the future. And I supplement the source of data by constructing the index system in the article. Through NVivo 11.qualitative analysis software, based on Grounded Theory, the policy text is encoded at three levels.

Thank you very much for your attention and time. Look forward to hearing from you.

Your sincerely,

Ningzhi Li,

School of Public Administration, China University of Geosciences, 388 Lumo Road, Wuhan, Hubei 430074, China

E-mail: [email protected];[email protected]

Round 2

Reviewer 1 Report

It is fine